# Improving spaces for women first responders: A grounded theory on gender equity

Kelly B. Gregory[1]*, John G. Mielke[2], Elena Neiterman[2]

**1** Dalla Lana School of Public Health, University of Toronto, Toronto, Ontario, Canada, **2** School of Public Health Sciences, University of Waterloo, Waterloo, Ontario, Canada

* kelly.gregory@mail.utoronto.ca

## Abstract

Emergency response work has historically been performed by men and thus designed with them in mind; however, during the past few decades, increasing numbers of women are conducting this work. Despite growing participation, research suggests women first responders continue to face unsupportive workplace structures and cultures. This study explored the occupational experiences of women who work as firefighters, police officers, and paramedics from Southern Ontario, Canada. Semi-structured interviews conducted with this population ($n = 20$) focused on resiliency and stress, diversity and inclusion, and gender and the role of professional identity. Constructivist grounded theory guided analysis and cross-profession comparisons. Participants described significant improvements to women's inclusion in first response work, however, they also identified continuing challenges. While some environments were described as highly supportive, many women still faced sexism and glass ceilings. Despite persisting obstacles, participants were deeply passionate about their work, and actively encouraged other women to join the field. Study results suggest that future advances can be encouraged by addressing the need for improved access to uniforms and equipment, on-the-job training to address barriers to promotions, flexible scheduling and childcare supports, and legislating equity, diversity, and inclusion training for all leaders and workers in the first responder community.

## Introduction

Women are said to offer unique skills to first response work, such as emotional intelligence, a lower likelihood of using excessive force, and the capacity to drive organizational change [1,2]. Despite these and other valuable attributes, women are substantially underrepresented in the field of first response work, and recruitment and retention continue to advance slowly [3,4]. Notably, we take "first response work" to include those professions that respond to and manage medical, safety, or

**Data availability statement:** Data cannot be shared publicly because of ethical reasons, as they contain identifying and sensitive information. Data are available from the University of Waterloo Institutional Data Access / Ethics Committee (contact via researchethics@uwaterloo.ca) for researchers who meet the criteria for access to confidential data.

**Funding:** Kelly B. Gregory received financial support to conduct this work from the Social Sciences and Humanities Research Council of Canada Joseph-Armand Bombardier Canada Graduate Scholarship (CGS-M), and the Department of National Defence through the Mobilizing Insights in Defence and Security (MINDS) scholarship initiative. Further information is available here: https://www.nserc-crsng.gc.ca/students-etudiants/pg-cs/cgsm-bescm_eng.asp Please note there is no grant number associated with these awards. The funder had no role in study design, data collection and analysis, decision to publish, or preparation of the manuscript.

**Competing interests:** NO authors have competing interests.

environmental crisis situations in community settings, such as those in the fire service, police service, and paramedicine.

The apparent underrepresentation is echoed in the literature, where knowledge about the experiences of women first responders is limited. However, women in firefighting have been found to face challenges related to inclusion, which can manifest as ill-fitting safety equipment [5], harassment, discrimination, and bullying [6], as well as various forms of tokenism [4,7–9]. These challenges have been attributed to the profession's historical adoption of a heavily masculinized archetype, which is understood to be galvanized with professional competency and described as a white, middle class, heterosexual man who is physically oriented, altruistic, and heroic [7,8,10,11]. As a result, workplace culture, which typically acts as a protective factor in demanding and stressful occupations, can become a barrier to belonging for women in firefighting [12,13].

Women working as police officers also appear to face "hypermasculine environments" [2,14], contributing to discrimination and harassment and psychosocial stress [2,15,16]. Hypermasculine attitudes within the police service can be produced both on the job and within training at police academies as a subtext of professional socialization [17], in addition, values such as "chasing the bad guys" [14] and an "old boys club" culture [13] can contribute to resisting the development of gender equity for women.

Notably, there is an even greater scarcity of literature on women in paramedicine, when compared to firefighting and policing, even though the percentage of women employed in paramedicine is greater than the combined percentage of women working in the police and fire services. Much of this research addresses the mental and psychological wellbeing of women who work as paramedics [18–22]. However, a Canadian survey [23] described women paramedics at an increase risk of intimidation (OR, 1.35; 95% CI, 1.07–1.70), physical assault (OR, 1.36; 95% CI, 1.06–1.74), sexual assault (OR, 5.45; 95% CI, 2.74–10.84), and sexual harassment (OR, 5.95; 95% CI, 4.31–8.21) from patients, families, and colleagues, compared to their male colleagues [23]. As well, women paramedics in Riyadh, Saudi Arabia, reported difficulties in balancing childcare and domestic work with variable shift work scheduling [24]. These participants also faced challenges surrounding physical fitness and strength, as well as a lack of workplace infrastructure to provide sufficient privacy [24]. In this context, women first responders were highly valued for their capacity to provide emergency services to women patients within the constraints of cultural and religious requirements, such as those surrounding bodily privacy between men and women [24].

Emergency response efforts across Canada are legislated at the federal, provincial/territorial, and municipal levels, and primarily center around the deployment of service providers through a triage and dispatch process, upon receipt of a 9-1-1 call, or text message [25]. While there are many first responder roles, including 911 dispatchers and emergency medical technicians, the scope of this project focuses on those most commonly found in the Canadian context: paramedics, firefighters, and police officers. The demographic composition of women who work as first responders

in Canada, at the time of writing, include 12,797 (36.5%) paramedics [26], 16,008 (23%) police officers [27], and just 293 (3.3%) career firefighters [28]. Collectively, these data suggest there are almost 30,000 women working as first responders in Canada, however this number is likely underestimated, as many provinces do not record these gendered data, nor do they include volunteer and forest firefighters. Critical shortages in first response work continue to persist across Canada, and while the proportion of women is growing, demographic data on gender distribution, professional occupations, and professional roles among Canadian first responders is disaggregated and inconsistent. Furthermore, data on women's experiences of working in these occupations remain sparse [13,28]. The present research is in response to this gap in the literature. The specific aims of this study were to explicate the experiences of women paramedics, police officers, and firefighters in Southern Ontario, Canada, and to advance gender equity within the field of first response by providing an evidence base that may support women's recruitment and retention.

## Methods

Upon receiving ethics clearance from a University of Waterloo Research Ethics Committee (ORE# 43443), recruitment involved outreach to key contacts in each profession, entailing a request to share a recruitment poster through internal communication channels that would reach workers. The inclusion criteria included individuals who self-identify as women, who were currently working in Southern Ontario as a police officer, firefighter, or paramedic, or were on leave or had recently retired within the last three years. The recruitment poster asked interested individuals to contact the lead researcher directly, so the desire to participate remained undisclosed to the original key contacts who may have held organizational authority over interested volunteers. Additionally, the recruitment poster was shared over social media, and snowball sampling was employed. The recruitment period ranged from October 1, 2021 to February 28, 2022. Informed consent was obtained through a process approved by the institutional Research Ethics Board where interested volunteers were emailed a letter of information about the research study and an informed consent document. To participate, interested individuals provided a signed copy of the informed consent document in advance of the interview. Participants were also given the option to provide consent orally, in which case, at the beginning of the interview, the lead author would read through the informed consent document and request that the participant respond verbally to the questions in the consent form, while being audio recorded. The lead author archived participant consent on a password-protected computer.

### Data generation and analysis

Our interview sample included paramedics ($n=7$), police officers ($n=8$), and firefighters ($n=5$) from Southern Ontario, Canada. The single-point interviews were conducted by the lead author either by phone ($n=4$), or Zoom ($n=16$), at the choice of the participant. Participants took part in interviews in their homes, at work behind closed doors, and in other undisclosed environments, and several participants were in the company of their young children. At the time of conducting interviews, the lead author identified as a woman who had previously worked in several men-dominated fields, and possessed academic training and professional experience in qualitative health research through an ongoing Master's of Science in Public Health degree program, both of which likely facilitated an ease of conversation with women first responders, and a heightened awareness of respecting the needs and boundaries of participants. There was no previous relationship between the research team and the participants prior to the study. The interview guide was developed by all three authors, and while was not tested prior to use, was reviewed by two additional members of the lead author's advisory committee. The guide consisted of both demographic and open-ended questions related to the individual life course of participants, resiliency and stress, workplace culture, diversity and inclusion, and gender and professional roles. (See Supplementary Material for interview guide.)

Demographic information provided context of participant lives during interviews and were used during recruitment to select the broadest range of participants possible across demographic and professional attributes (e.g., age, race, sexual orientation, profession, rank, location). Interviews, which lasted from 53 to 167 minutes, were audio-recorded and

transcribed using transcription software, and transcripts were reviewed for accuracy. All recordings were subsequently deleted. Participants who conducted their interviews over Zoom were given the option of turning their cameras off. In the instances where cameras were on, video recording was captured, however visual data were not directly analyzed (e.g., body language, or gestures) and video recordings were protected and ultimately deleted in the same manner as the audio recordings. Memos were written following each interview and reviewed during analysis, where the coding, organization, and conceptualization of the data were conducted with NVivo Pro 12 software. At twenty interviews, enrollment was stopped given the richness and diversity of the interview data and that a repetition of narratives was beginning to occur.

Constructivist grounded theory [29] was used during analysis to generate a descriptive and interpretive theory of women's experiences as first responders. In alignment with this approach, data collection and analysis were conducted in tandem, to facilitate an iterative relationship, and to develop theory which could be continually tested against all the data generated [30]. Inductive line-by-line coding was conducted by the first author with the first two transcripts. The names of codes were largely drawn from the language of participants in the data. After reviewing with the second and third authors, the codes were collapsed into focused codes and analytic categories, forming components of an explanatory narrative. This level of categorization was driven by the specific aims of the study and informed by the trends and contradictions in the data. The remainder of the transcripts were subsequently coded with the ensuing codebook, and new focused codes were added when novel topics arose. Additionally, the initial two transcripts were reviewed again at the end of the coding phase, to see if any of the new codes generated were relevant. A plain language summary and a report of the findings were shared with participants to provide them with the opportunity to make comments, or edits prior to completion of the study.

## Results

### Study population

Among the ($n = 20$) participants in this study, the average age was 39 years old (range = 27–58) and six participants identified as a sexual minority (e.g., bisexual, lesbian, gay, hetero-flexible). There was a diversity of family status among participants, with nine married, or partnered with children, four participants separated or single with children, four single or divorced, and three married/partnered without children. All participants attended some form of post-secondary education (college, $n = 7$; university, $n = 10$; graduate degree, $n = 3$) and all identified as "female" gendered, English-speaking, and white. Participants worked at a variety of ranks ranging from probationary roles to senior level management, with an average length of service of 13.25 years (range = 0.38–37 years). Of the participants, thirteen identified as working in urban locations, two in rural locations, and 5 in mixed locations.

### Findings

The following section presents a collection of four themes drawn from the data: changing professional environments, impacts of performing womanhood, facing resistance, and shaping and constructing spaces for women. These four ideas capture various aspects of participant experiences as first responders, encompassing individual actions, organizational structures, and transitions across time.

### Changing professional environments

First responder organizations continue to undergo significant transitions related to gender equity. While many participants described that very few women hold leadership and upper-management positions, those with longer service records shared how many more women colleagues they now have compared to when they began their careers. Some participants also spoke about increasing accessibility to uniforms and equipment designed for women's bodies. For example, the paramedic's stair chair with longer arms and a woman's MOLLE vest (a tactical vest worn by police officers) are designed to provide more space for breasts in their use, or, in the case of firefighting, smaller oxygen face masks that accommodate

a women's typically more narrow face shape. Participants also described how the standards for accomplishing work tasks were becoming more oriented toward safety for various body shapes and sizes and mentioned the introduction of training programs that teach women how to leverage their specific body mechanics. As a result of the increasing development and use of technology, participants described how there was becoming less need for extreme strength and size, when compared to historical standards.

Changes to workplace cultures were also noted, such as a reduction in the often-casual sexist commentary, or harassment that regularly took place. Expectations for the types of tasks women perform, and how they are carried out, also appears to be changing. For example, several paramedics described that male firefighters used to routinely perform all the lifting work on an emergency scene when women paramedics were present:

*"So, yeah, …the firefighters were the first kind of, like, eye opening experience, like wow, they really think I can't do this JUST because I'm a woman. One hundred percent.." - PM-03*

Thanks to the development of mechanical stretchers, as well as the insistence by women that they are capable of lifting, this paramedic described this presumption as generally no longer the case.

In addition to shaping day-to-day experiences, workplace cultures can also communicate the values of an organization through the presence, absence, or the specific impacts of women-focused policies. Police officers and firefighters recounted that, historically, when women disclosed a pregnancy to their organization, they would be immediately pulled from public-facing positions and reassigned to, often dreaded, administrative roles. Some police officers described a current practice in which a pregnant officer must relinquish their "use of force" privileges (the legal ability to use physical effort to control, restrain, or overcome the resistance of another person) when they no longer fit into their policing uniform. However, in most cases, participants described that women now have relatively more autonomy over how long into their pregnancy they might remain in a role, and how re-deployments are often lateral, or even upward professional opportunities.

Participants also described the trajectory of their individual resistance to harassment and discrimination, through an increasing willingness to follow up on and address sexist comments over the course of their career. Several participants also connected this willingness to their career stage and level of self-confidence, such as the police officer quoted here, who worked for the service for more than a decade:

*"It's interesting to go through your career and until you feel comfortable to stand up, until you have enough, um, confidence within yourself and your abilities to actually stand up to people like that, it's tough."- PO-03*

As alluded to by this police officer, many participants described enduring harassment and discrimination in the early parts of their careers. However, through continuing to develop and succeed professionally, they increasingly became more empowered, personally and professionally, to shape work policies, practices, and cultures into spaces that were supportive for women.

When reflecting on the stories that participants shared, it was clear that women appreciated the organizational developments that have been accruing over the last few decades to accommodate women's bodies and unique needs, as this police officer describes:

*"…and I kind of hope, and feel it's better today. Like policing today is not the same as it was 20 years ago. And yet, there are still little snippets and little pieces here and there that I see, that I think, maaan, have we really come that far? Or are we still in exactly the same spot we were 20 years ago?" – PO-02*

While participants, such as this officer, were happy to report increasing numbers of women colleagues, and improving access to resources and technologies, they also agreed that there is still much room for improvement.

## Impacts of performing womanhood

Participants in this study described how being a woman can be an asset in some situations, but remains a liability in many others. Communication and de-escalation skills, commonly seen as abilities that are intrinsic to womanhood, were described as highly regarded by colleagues and the participants themselves. A firefighter who worked in a large city service described the value of connecting to clients and taking flexible approaches to problem-solving, which she attributes to the gender scripts of being a woman in the following way:

> *"I would say, in general, our bedside manner is quite good. Just because …like we just need to communicate little bit better. Maybe naturally a little bit more nurturing, a little bit more patient and understanding, a little bit more empathetic, sympathetic,… I think our character is just, you know, not everyone is maternal and paternal and nurturing and whatever. But I think, in general… we seem to establish a connection with our patients a little bit more… I see that, like, ability to process really quickly, evaluate really quickly, create… an approach, and then switch from plan A to plan B, to plan C, as needed. Where maybe our (male) counterparts are just a little bit more tunnel vision." – FF-01*

Participants clearly valued these abilities and felt they positioned women as experts in this work, even at times having an advantage over male colleagues, as this police officer describes in the context of handling a high-intensity call:

> *"…speaking from my experience, women are better communicators. We can do the job without it becoming physical. Nine times out of ten. Ninety-nine times out of one hundred. And I've been to calls where, you've got the call in hand and your male counterpart shows up and all of a sudden you're in a fight and you have no idea how that happened. So you know, I think in a lot of ways women bring different things to the job than men do. I know, (in) hand to hand combat with a guy, I'm gonna lose. That's why I have a tool belt. That's why I have use of force options. But that's primarily, my communication skills, are what's going to keep me out of those situations."- PO-06*

As participants listed these strengths, they also described how having a differently shaped body can introduce further challenges to their work. Some individuals described having no issues with fitting into uniforms, which they explained was due to their body being larger, wider, and taller than the average woman's body. Yet many noted feeling unprofessional, or unsafe when having no other choice but to wear poorly fitting uniforms, and some described needing to make personal alterations to uniforms when pregnant.

While some participants discussed convoluted, or undefined processes for requesting maternity leave, negotiating the demands of work and home life was clearly a burden for many of the women interviewed. However, as a firefighter who worked in a larger organization observed, the presence of children at home was not the sole factor generating stress related to work-life balance:

> *"Yeah, I know, like, I work with a crew with, you know, all the guys have young kids, and it's a break for them, coming to the hall, because it's like, their lives are definitely, you know, difficult. But I do actually know a few women who, had a baby, and, have been off on mat leave for quite a few years. And it's very hard to manage all of it." – FF-05*

As this firefighter clarifies, the stress of work-life balance was a gendered experience. While this participant felt that work can help to balance home-life stress for men with children, she described the same situation for women as being additive, or accumulative. Among all of our participants who cared for children, each one described how challenging it was working variable shift work schedules, such as this paramedic whose partner also worked a rotating shift schedule:

> *"…when I first went back to work with my daughter, I put her into daycare, because what else do you do? And I just felt like such a garbage mom, because… she'd just be there for like 12 or 13 hours, like goodness!"*
>
> *– PM-04*

Many participants explained how they struggled to navigate childcare and some suggested organizing pooled daycare services for first responders. However, participants often stated, particularly those in policing, that they would struggle to trust this kind of service. The hesitation was explained as being related to trusting the individuals doing the care work, or that it was considered risky to gather the children of many police officers in one location, for fear of heightening the visibility, or vulnerability, of their children.

## Facing resistance

Many participants discussed barriers related to the equipment and the environments in which they performed their work. A volunteer firefighter described her luck in having access to backup bunker gear:

> *"I was lucky enough that there was a set of spare gear that fits me perfectly. But had that gear not been there… Would I have been in a disadvantage? Absolutely. Because it's all huge men sizes. I would- it, it would have been a nightmare." – FF-04*

The presence of this backup gear being attributed to luck, rather than to sufficient planning and organization, speaks to the daily challenges that some women continue to face at work that de-prioritizes women's safety and inclusion. Access to washrooms at the scene of an emergency was also a common challenge for women firefighters, who described strategies such as bringing urination tools that allow women to expel urine while standing, as described by the following firefighter:

> *So they make disposable ones that are made of cardboard, which is great, because I'm not trying to put a pissy funnel back in my bag and have it like, touch my things. …and then I'm going to get back to the hall and wash it in the bathroom?! Like it's just- we'll let it dry?! So I just, they have disposable ones. So I buy the disposable ones. Because again, you're at a fire scene for hours. And you have to hydrate. If you do not drink 20 bottles of water, you're gonna pass out, because it's SO hot. It is 400 degrees. And you're wearing a giant oven mitt. …and you can't take it off. First of all, it's got suspenders. So, now I have to take my f---ing jacket off. And then the suspenders. And then like the boots are up to- it's just, it's a nightmare. I'm not- it's not gonna happen, especially in the winter." – FF-04*

Despite the combination of limited environmental supports in addition to gendered uniform design, this firefighter explains how women find strategies to cope. However, in this professional context, it was noted that if a firefighter's uniform becomes soiled with carcinogens, hazardous materials, or biological contaminants, it must be sent away to be professionally cleaned. Some of the firefighters interviewed described having access to only one set of bunker gear, and shared that if it becomes soiled with menstrual blood, they would spend time spot-cleaning their uniforms, rather than lose their gear, and thus their ability to perform their work.

Environmental factors also posed challenges for women who work as police officers, such as needing to remove bodily-worn gear, like a gun belt, and finding a sanitary place for its temporary location while going to the washroom: *"Where does it go? On the floor?!"* (PO-08). The work of policing can also require remaining in a stakeout vehicle for extended periods of time. In these cases, women described how challenging it was to urinate into a bottle, or other vessel, as their male colleagues did, and, in several instances, women described simply not applying to positions that require this type of work, given that this was a known expectation accompanying the role.

Women also described experiencing a social resistance to their presence in the workplace and the performance of their work. Women paramedics described regularly being scrutinized by patients who vocally question their skills and abilities (*"Are YOU gonna lift me?!" - PM-01)* and some of the police officers we interviewed shared similar experiences.

> *"So I might show up on scene… most men that I deal with are respectful, and, you know, appreciate my being there and listen to what I have to say. But, you know, when you get those older men who've been around a while, they*

*don't- they DO NOT like to be told what to do. So, I definitely have to kind of tiptoe on how I- I initially will tip toe, and if they don't listen to what I need, then I will have to kind of escalate it." – PM-06*

Some participants also noted a culture of specific language being used to define women as separate from the rest of their workforce, as noted by this high-ranking police officer:

*"…one of the terms that, when I first started was popular was 'PW', which is 'police woman'. And I hate that phrase. And some of the men that I work with now, even still, who have been around for 20 plus years, use it. And I will say to them every time, please don't call it that, like, 'police officer'. That's all you need to say. You don't need to differentiate between police woman and police man. Because oftentimes when the guys would say PW, it was derogatory. It was, oh here comes another PW." – PO-08*

From this perspective, it is clear that male hegemony is continuing to support workplace cultures that differentiate and subordinate women police officers from their male colleagues. Furthermore, job tasks that were particularly associated with masculine traits, and yet otherwise standard performance expectations, seemed to garner women special attention. This constable described receiving accolades for getting physical with someone during an arrest:

*"You would think that that would be, uplifting. But it was completely condescending to me… I will never forget that, my entire career. And I, I even think he meant well. You know, I think he was trying to pump my tires. I think he was trying to, you know, say 'good job'. But, that was not something any male officer ever would have been congratulated for doing. So, it's always that kind of a double-edged sword, right? Like you're getting a compliment, but it's a backhanded compliment." – PO-02*

As this officer appreciates that the comments may be innocent and even well intentioned, she reflects on a gesture which reinforces the relationship between gender and performance standards, as one that echoes in her memory as deeply insulting.

In addition to being challenged by their co-workers, participants also described resistance from other first response professions as well. In discussing the role of an Advanced Care Paramedic, this participant discusses delegating tasks on an emergency scene, and how gender shapes the way her actions and identity are perceived:

*"…as a female advanced care paramedic, you got to kind of be, at least I try to be- I don't want to say delicate, but I'm always wary of how I come across to my peers. Because I don't want to come across as a bossy bitch, right? Unfortunately, if I were a man coming in and saying the same words, in the same tone as I am, I'm now conceived, I'm perceived as bossy bitch. And they are perceived as, wow they're doing a great job taking (control). They're a LEADER!" – PM-02*

In response to resistance, participants expressed concern for their professional relationships and for their reputations, and shared the need to challenge and defend their suitability and merit when applying for and garnering promotions. In turn, participants described walking on eggshells when giving direction to their staff and employing participatory leadership styles as a tactic for reducing confrontation in their daily work. Furthermore, women who were navigating career advancement in conjunction with pregnancy and maternity leave, described taking on training during maternity leaves, or returning from maternity leaves prematurely *("you're constantly feeling like […] you're trying your best to prove yourself and to prove that work is a commitment as well. So, I actually came back early from mat leave." - PO-08)* in order to ensure promotional opportunities were not missed, or to guard against the risk of being perceived as someone who does not care about their career.

Amid increasing pressures to improve the number of women in leadership roles, participants described equity-driven promotions as helpful in addressing vertical gender distributions across organizational ranks. However, these promotions have come with consequences that have posed a threat to women's reputations. Across all professions, participants described how women's promotions have been contaminated with the possibility that they are unfounded and unearned, and simply a result of gender distribution-based decision-making. Participants noted that individuals of all genders are becoming increasingly skeptical of women in leadership roles, believing in a lack of merit as a baseline, until women prove themselves otherwise. Participants who garnered promotions also disclosed they experienced a quiet, though often deep, sense of self-doubt, questioning whether to attribute their career advancement to their skills and abilities, or simply to their gender.

*"Like, I don't want to stand out as a visible minority or being part of a group, to get anything, ever. And so, those two kind of comments or situations, (sighs) tainted the (promotion) process a little bit for me. You know, I'm still super happy and proud to be where I am. But, there's just that little niggling feeling at the back. Was it on me? Or is it because I'm a woman?" (PO-06)*

In the context of policing, this situation is further shaped by the public nature of the profession. Many police officers described observing equity-based promotions, where they felt women were promoted without the necessary skills and abilities, and described watching these women inevitably fail, both professionally and publicly. Furthermore, the increasing personal accountability and professional risk in conducting police work was identified as contributing to a decreasing status of policing as a "good job", particularly for women.

When considering the stories of women across all three professions, it became clear that being socially included is deeply important for one's mental health, one's access to resources such as mentorship, and to career development success. The degree of significance and importance of these factors is equal to the degree that women are willing to challenge the resistance they face, and accept risks as individuals and professionals, in order to drive gender and cultural advancements in their field.

## Shaping and constructing spaces for women

Despite being faced with enduring and multifaceted challenges, many participants repeatedly described how much they loved the work they do, and the teams of people they do it with. Participants described feeling part of a fraternity and a brotherhood that was strengthened by the shared trauma inherent in first response work. Many participants appreciated how the history of women's inclusion into their teams has not developed evenly across the field, with some noting their particular department as exceptionally invested in their success.

When probed further about her experiences of support, this firefighter described the reactions that her male colleagues expressed when recognizing one of their teammates is not being treated well:

*"One part that I do find interesting is like, my male counterparts gasping, like, 'DID THAT PERSON REALLY JUST SAY THAT?' Because they know me as me. And they're just like, oh my gosh, like, I never thought that you girls would have to deal with that. Like, they're just like, gobsmacked by it. Because, you know, they understand our capabilities. And that we're here for a reason. But like when they hear it second hand or, or like, you know, hear it from our point of view, they're just like, they're baffled. Which, which is, is kind of good, because it's like, yeah, this, this is how it goes often." – FF-03*

Other participants described how their male colleagues would offer support, (*"Do you want me to go talk to that guy?!"* FF-04) when observing poor behaviour, and women recognized these gestures as proof of a genuine alliance across genders.

When discussing their careers, women often explained how they cannot imagine themselves doing anything else, and how much they wished to encourage other women to join them. As women appreciate what they have, they also recognized the work that remains. While some participants described difficult relationships with their female co-workers, most discussed the positive experiences of mentorship they had with women leaders, who provided them with examples of what is possible for women first responders. As this primary care paramedic suggests, many of the women first responders seek to mentor and defend one another when needed, and to lead by example with strength and perseverance:

> "So there definitely are still issues between what women will have (to face) in the workplace that men probably don't, just based on a vulnerability thing and society always thinking that women are supposed to put up with this [expletive]. But I like to think that I've become the person who makes that stop and becomes a good example for other women to feel strong and supported as well. So that's how I feel- like I'm trying to be better, and then also kind of represent that to other people as well." – PM-01

Participants specifically named the role of women trailblazers, who were the only women on their teams at each step of their career, and are now in professional positions of power, as those who are making impactful change at the organizational level for other women. Participants described that the more women reach down to pull other women up through the ranks, the more lower-ranking women reach up for help to achieve their goals.

During interviews, participants were asked to identify their favourite part of their job, in the interest of understanding how to strengthen recruitment and retention strategies for women first responders. Women often cited the high-adrenaline moments of driving at high speeds with lights and sirens, and fighting injustices in defence of their communities. Participants also explained how their ever-changing work environments meant they were never bored, and the sense of power they felt when genuinely helping someone through a crisis. Participants often described feeling deeply connected to their communities and being nourished by the constant intellectual challenges in their work.

## Discussion

In several instances, our findings align with the literature, much of which identifies the challenges related to workplace cultures and policies, or practices that fail to support women's specific needs [12, see also 31]. The various forms of harassment and discrimination, as well as tokenism, discussed in prior research continue to persist for many women first responders [4, 7-9]. Likewise, problems accessing uniforms and equipment designed specifically for women, as discussed by Griffith et al. [5], remained a challenge for some of the participants in this study.

Alternatively, the existence of hypermasculine work environments, addressed by many experts in the field [2,8,10,11,14,17], appears to be shifting in some circumstances. Changes to work environments were explicitly noted by women who were working under the supervision and management of other women and exceptional men. Additionally, as organizations adapt and respond to public opinion, or patient needs, they appear to be increasingly shifting towards typically-feminine values which will shape workplace cultures to further appreciate the skills at which women often excel.

This study provides novel substantive and methodological contributions to the literature through its focus on and between three professions, improving understanding of both the complexities and larger trends in the field of first response work. These approaches have highlighted the differences between the professions, such as training and advancement practices, which in the case of paramedicine may be contributing to barriers for women aspiring to leadership roles. Our work has also articulated some of the unique factors shaping workplace cultures in each profession, such as the influence of public opinion in policing and firefighting, or improving access to equipment and technologies in paramedicine. This work has also examined the gendered nature of each occupation, such as the feminine-coded care work of paramedicine, or the masculinized heroic perceptions of firefighting, contributing further insight into persistent gender distributions across first response professions.

Our research has also made unique contributions to the exploration of generational differences and how they intersect with gender and rank, and uniquely presents positive participant stories, which describe women feeling embraced by their teams and genuinely belonging to their fraternities. Finally, this study has contributed a novel perspective through its comparison of three first response professions, reflecting healthcare, environmental protection, and public safety, as they work individually and collaboratively to serve the general public in their times of need.

## Recruitment and retention recommendations

Many of the interview questions asked of our participants were relevant to the recruitment and retention of women in first response professions. The following summary represents an analysis of the data with this specific lens, and is also summarized in the list below.

In response to hearing from women about persisting challenges related to uniforms and equipment, it is recommended that organizations ensure their women first responders have ready access to gear that fits well and ensures safety and ease of use. Furthermore, it is recommended that all organizations prioritize the provision of multiple sets of uniforms to ensure women do not have to shoulder the work of extra cleaning, and to remove potential barriers for women to fully participate in their work.

Participants also recognized the importance of increasing the number of women first responders in leadership roles. In particular, providing on-the-job paid training to facilitate advancement for women paramedics may help to dismantle the glass ceiling described by participants in this study.

There were also calls for increased support from management for the periods during and after which promotions take place. It was repeatedly communicated that women only want to be promoted when they have genuinely earned the advancement. As such, organizations must work to combat the negative perception that women are gaining from equity-driven promotions through transparency in hiring processes and the provision of ongoing training and mentorship to exemplify clear support for women's professional success.

In light of this study's findings, women first responders would likely benefit from supports which ease tensions between work and home life. Such supports could take the form of formal childcare with flexible options, such as professional daycare and personal childcare services. Creating scheduling approaches that provide workers more opportunity to shape their regular schedules would also support women who care for others. For example, supportive schedules could be enacted through adjustments to the specific hours worked in day, or which days are taken off, as well as offering more options around the number of hours worked across a week.

Lastly, participants called for continued efforts to improve workplace cultures to create spaces that are positive and supportive for women to conduct first response work. Women identified the importance of delivering equity, diversity, and inclusion training for leaders and workers; a program which could be mandated by provincial legislation for all professional levels and geographic regions.

## Limitations

The empirical findings of this study are geographically, historically, and socio-politically situated within the southern region of the province of Ontario, Canada. However, this study may offer conceptual and substantive relevance to other geographic and professional jurisdictions, both nationally and globally, who face similar circumstances.

Limitations do exist surrounding the choice to focus on women's experiences. Gay men, men who sleep with men, or non-binary and transgender individuals are likely to be impacted by gender norms and expectations in these occupational settings [4,8,32]. While their inclusion was beyond the scope of this study, our findings point to the need for further research with these populations.

Furthermore, according to their self-disclosure, our participant group was entirely composed of white, English-speaking, cis-gendered women. This has certainly influenced the type of responses and experiences shared with this study, where

varying degrees of privilege through whiteness, language, and normative genders will have shaped these experiences. As such, our results do not represent the experiences of all women, which highlights other important avenues for future research to investigate.

Lastly, given that we focused on the first responder groups which make up the majority of those who work in the Southern Ontario context, we were unable to investigate the experiences of women who work in first response professions beyond the three previously mentioned. As a result, and in line with the dearth of literature on the subject, future research is required to gain a sense of gender equity in other first response roles, such as Emergency Medical Technicians and 911 dispatchers.

## Conclusion

The primary goal of this investigation was to explore and articulate the experiences of women first responders. In doing so, we have also shed light on the factors shaping these experiences. Findings have been organized and understood through the development of four themes: (1) changing professional environments, (2) impacts of performing womanhood, (3) facing resistance, and (4) shaping and constructing spaces for women. When taken together, these themes connect to form a theory about women's experiences, which encompasses their accounts of change, struggle, success, and growth. This theory suggests that the significant improvements to women's inclusion in first response work have distributed unevenly across the field. As such, while some women are working in highly supportive environments, many are still facing sexism and glass ceilings. Despite this challenge, women are deeply passionate about the work they perform, and actively encourage other women to join them. This theory speaks to each of the professions individually, as each is experiencing its own trajectory of development, yet it also holds relevance to the collection of women first responders in Ontario, who perform these vital services for members of the public.

Considering the findings by profession allows for comparative insights to be gained. In the cases of paramedicine and police services, where more women have been part of the workforce for longer periods of time, more space has been created in their organizations for women's perspectives and approaches. Organizations with higher ratios of women tend to have a greater need, as well as a justification, for identifying adaptive strategies, or for providing supportive technologies for women to succeed in their roles. In these cases, the organizations are clearly doing the work of adapting to the women they employ. Alternatively, in many fire departments, there remain very few women at all organizational levels. This circumstance positions individual women to take on the work of adapting to their environments, in which they must become "one of the guys" to be successful in their work. As a result, some participants actively challenged the heavily masculinized archetype of the first responder [7,8,10,11], while in other cases, they strove to align themselves with the dominant characteristics of their field. Taken together, it is clear that, while women find ways to leverage their skills and abilities for success, they continue to navigate colleagues, environments, and cultures which challenge their participation in first response work.

## Supporting information

**S1 File. Interview Guide.**
(PDF)

**S2 File. Key recommendations for improving spaces for women first responders.**
(DOCX)

## Acknowledgments

The original thesis from which this article derives is publicly available through the University of Waterloo thesis repository. We would like to thank the participants of this study for sharing their time and energy, and their deeply personal experiences, as well as numerous first response organizations who facilitated some of our recruitment strategies. We would also

like to thank Dr. Meg Gibson and Dr. Phil Bigelow for their valuable insights during the development and the conducting of this study.

## Author contributions

**Conceptualization:** Kelly B. Gregory, John G. Mielke, Elena Neiterman.

**Formal analysis:** Kelly B. Gregory.

**Investigation:** Kelly B. Gregory.

**Project administration:** Kelly B. Gregory.

**Supervision:** John G. Mielke, Elena Neiterman.

**Writing – original draft:** Kelly B. Gregory.

**Writing – review & editing:** Kelly B. Gregory, John G. Mielke, Elena Neiterman.

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
