## [Decision Letter · Decision Letter 0]

25 Mar 2025

PONE-D-24-42534
Improving Spaces for Women First Responders: A grounded theory on gender equity
PLOS ONE

Dear Dr. Gregory,

Thank you for submitting your manuscript to PLOS ONE. After careful consideration, we feel that it has merit but does not fully meet PLOS ONE’s publication criteria as it currently stands. Therefore, we invite you to submit a revised version of the manuscript that addresses the points raised during the review process.

We look forward to receiving your revised manuscript.

Kind regards,

Saravana Kumar

Academic Editor

PLOS ONE

Journal Requirements:

2. In the ethics statement in the Methods, you have specified that verbal consent was obtained. Please provide additional details regarding how this consent was documented and witnessed, and state whether this was approved by the IRB.

“Kelly B. Gregory received financial support to conduct this work from the Social Sciences and Humanities Research Council of Canada, the Joseph-Armand Bombardier Canada Graduate Scholarship (CGS-M), and the Department of National Defence through the Mobilizing Insights in Defence and Security (MINDS) scholarship initiative. Please note there is no grant number associated with these awards.”

Before we proceed with your manuscript, please address the following prompts

5. We note that your Data Availability Statement is currently as follows: [All relevant data are within the manuscript and its Supporting Information files.]

6. Please ensure that you include a title page within your main document. You should list all authors and all affiliations as per our author instructions and clearly indicate the corresponding author.

7. Please update your submission to use the PLOS LaTeX template. The template and more information on our requirements for LaTeX submissions can be found at http://journals.plos.org/plosone/s/latex.

8. Please include your tables as part of your main manuscript and remove the individual files. Please note that supplementary tables (should remain/ be uploaded) as separate "supporting information" files

Additional Editor Comments (if provided):

Comments from PLOS Editorial Office: We note that one or more reviewers has recommended that you cite specific previously published works. As always, we recommend that you please review and evaluate the requested works to determine whether they are relevant and should be cited. It is not a requirement to cite these works. We appreciate your attention to this request.

Reviewers' comments:

Reviewer's Responses to Questions

**Comments to the Author**

1. Is the manuscript technically sound, and do the data support the conclusions?

Reviewer #1: Partly

Reviewer #2: Yes

2. Has the statistical analysis been performed appropriately and rigorously? 

Reviewer #1: Yes

Reviewer #2: N/A

3. Have the authors made all data underlying the findings in their manuscript fully available?

Reviewer #1: Yes

Reviewer #2: No

4. Is the manuscript presented in an intelligible fashion and written in standard English?

Reviewer #1: Yes

Reviewer #2: Yes

5. Review Comments to the Author

Reviewer #1: Abstract

Comment 1, Page 4, Line 1: Consider rephrasing “designed”. Emergency response work is not necessarily gendered in its design.

Introduction

Comment 1: Please be consistent with your language, public safety vs. first responder.

Comment 2: A definition of what a first responder is would be helpful to focus the reader on the cohorts your are specifically commenting on. There is a lot of back and forth in the introduction between FF, PO etc, so it would be help to define the groups you are presenting on.

Comment 3: It is worth mentioning the increased risk for adverse mental health outcomes among these groups, of which are further elevated among women. EMS and fire, specifically, are noted to have increase depression, anxiety, substance abuse, and even suicide. This speaks volumes when it comes to advocating for more research among these cohorts. Please consider including some of the citations below.

McCann-Pineo M, Keating M, McEvoy T, Schwartz M, Schwartz RM, Washko J, Wuestman E, Berkowitz J. The Female Emergency Medical Services Experience: A Mixed Methods Study. Prehospital Emergency Care. 2024 May 18;28(4):626-34.

Huang G, Chu H, Chen R, Liu D, Banda KJ, O’Brien AP, Jen HJ, Chiang KJ, Chiou JF, Chou KR. Prevalence of depression, anxiety, and stress among first responders for medical emergencies during COVID-19 pandemic: A meta-analysis. Journal of global health. 2022;12.

Stanley IH, Hom MA, Joiner TE. A systematic review of suicidal thoughts and behaviors among police officers, firefighters, EMTs, and paramedics. Clinical psychology review. 2016 Mar 1;44:25-44.

Stanley IH, Boffa JW, Smith LJ, Tran JK, Schmidt NB, Joiner TE, Vujanovic AA. Occupational stress and suicidality among firefighters: Examining the buffering role of distress tolerance. Psychiatry research. 2018 Aug 1;266:90-6.

Lebeaut A, Tran JK, Vujanovic AA. Posttraumatic stress, alcohol use severity, and alcohol use motives among firefighters: The role of anxiety sensitivity. Addictive Behaviors. 2020 Jul 1;106:106353.

Comment 4: What was the author’s justification for not including EMT’s? The literature suggest that they experience significant occupational and mental health burdens alongside their paramedic counterparts. There needs to be a firm rationale for their exclusion.

Comment 5: Has there been any efforts to collect the demographic distribution among the professions? What is the reason for not collecting it? Time, funding? Might be something to comment on.

Methods

Comment 1: Please refer to and utilize the Consolidated Criteria for Reporting Qualitative Research (COREQ) Checklist., There are numerous items that are missing and not included in the main text.

More detail is need on the recruitment methodology: how many agencies/institutions were provided with the recruitment poster? Besides being active/on leave/retired <3 years, was there any other inclusion/exclusion criteria?

Who developed the interview guide? Was it reviewed or tested prior to use?

Were the ZOOM sessions just audio recorded, or video as well? If video was used, please state. Further, if video was used, do you think that responses among those who opted for video/ZOOM may have been biased? Less confidentiality with video, which may have prevented participants from being 100% forthcoming.

Please indicate how long each interview session was.

There is no justification for the sample. Was saturation among themes met at 20? How did the authors decide to stop enrollment/interviews?

Who was responsible for the coding of the transcripts? How do you think the author’s own personal experiences may have influenced the coding? Reflexivity in qualitative research has become increasingly important.

Results:

Comment 1: are you able to present the demographic information by profession? This would be helpful for the readers. Further, is there any information on the EMS/fire agency type? Such as EMS agency vs. Fire agency with EMS? This distinction is meaningful and can directly impact participant experiences.

Comment 2: The presentation of the four themes is less than to be desired.

Firstly, it appears that there many subthemes within each of the overall themes. For example, ‘Changing Professional Environments’ appears to have subthemes of increases accessibility (uniforms, stair chairs, mobile stretchers, etc), workplace culture (sexism, harassment, women focused policies. This reviewer suggests that you consider reorganizing the presentation of the data to group quotes and ideas by subthemes. It is disjointed as it is currently presented.

Secondly, it may help focus and streamline things if you put all participant quotes within one large table. This would also cut down on the word count.

Comment 3: There are numerous occasions where expletives are included in the participant quotes. This review understands “fidelity” of the quotations; however, it is suggested to remove the curse words and replace with something like [expletive].

Comment 4: Under the ‘Facing resistance” theme, additional participant quotations are need to support the subthemes presented. Specifically, quotes commenting on pregnancy/maternity leave in its relation to career advancements, promotions (particularly being unearned), and social acceptance are needed. These are large paragraphs of text with no supporting quotation.

Discussion:

Comment 1: Please include Contursi et al 2018 as an in-text citation.

Comment 2: As mentioned previously, presenting demographic data by profession is necessary to support the claims in your last paragraph of the Discussion.

Comment 3: While this Review appreciates and agrees with the recruitment and retention recommendations presented in Table 1, many of them are anticipated to have low feasibility, particularly among EMS and EMS/Fire agencies with limited funding (i.e rural). At least within the US, the pay discrepancies across the three professions are profound—with EMS receiving a fraction of the salaries of their law and fire counterparts. This generates significant difficulties in instituting any organizational changes that would benefit women directly (i.e scheduling, childcare, etc). Perhaps providing some information on how the Canadian EMS system compares to other major nations would be helpful. Or, maybe provide an example as how the authors envision implementing these recommendations? On paper they are great, but to anyone in operations would see many of these as seemingly impossible to enact.

Secondly, perhaps separating the key recommendations by profession is also needed. Yes, all these recommendations will benefit women in each of the 3 professions, however, there are others that could be specifically tailored for EMS vs Fire vs. Law.

Limitations:

Comment 1: This may no longer be necessary after addressing the comment about saturation, but if saturation was not met this needs to be included as a limitation

Conclusion

There is no conclusion section. Please include.

Reviewer #2: This study set out what it aimed to do: to analyze information from women first responders and develop ideas to advance gender equity in the professions. Study significance is supported by the cited literature and using grounded theory to generate new knowledge was effective. The description of the grounded theory process was succinct.

A “major” issue is the abrupt ending of the paper. I wonder if the “Recruitment & Retention” and “Limitations” sections could come earlier in the Discussion section, so there can be a conclusion to wrap up the paper.

For clarity, restate the total number of participants in the “Results” section as you begin to describe the population sample. Additionally, defining the term “sexual minority” for your reader would be helpful.

Diving into the topic of first responders who identify as women is important in supporting the increased number of them in these historically masculine professions. I support the publication of this article with the suggested revisions above. Thank you for the opportunity to submit this review.

6. PLOS authors have the option to publish the peer review history of their article (what does this mean?). If published, this will include your full peer review and any attached files.

Reviewer #1: No

Reviewer #2: No

---

## [Author Response · Author response to Decision Letter 1]

11 Jun 2025

Response to Reviewers

PONE-D-24-42534

Improving Spaces for Women First Responders: A grounded theory on gender equity

PLOS ONE

We thank both the editor and the two reviewers for their consideration of our manuscript. We have carefully reflected upon the generous comments provided and have outlined below our responses to the concerns that were raised. We are hopeful that our manuscript will now be considered acceptable for publication in PLOS ONE.

Journal Requirements:

1. Please ensure that your manuscript meets PLOS ONE's style requirements, including those for file naming. The PLOS ONE style templates can be found at https://journals.plos.org/plosone/s/file?id=wjVg/PLOSOne_formatting_sample_main_body.pdf and https://journals.plos.org/plosone/s/file?id=ba62/PLOSOne_formatting_sample_title_authors_affiliations.pdf.

We have separated the submission into two files, as per the journal requirements: 1) Cover Letter; 2) Title Page and Manuscript. I have also adapted their formatting to align with the journal requirements, and have uploaded the interview guide as Supplementary Material.

2. In the ethics statement in the Methods, you have specified that verbal consent was obtained. Please provide additional details regarding how this consent was documented and witnessed, and state whether this was approved by the IRB.

We have elaborated on this process on line 153, so that the statement now reads as follows:

“Informed consent was obtained through a process approved by the institutional Research Ethics Board where interested volunteers were emailed a letter of information about the research study and an informed consent document. To participate, interested individuals provided a signed copy of the informed consent document in advance of the interview. Participants were also given the option to provide consent orally, in which case, at the beginning of the interview, the lead author would read through the informed consent document and request that the participant respond verbally to the questions in the consent form, while being audio recorded. The lead author archived participant consent on a password-protected computer.”

“Kelly B. Gregory received financial support to conduct this work from the Social Sciences and Humanities Research Council of Canada, the Joseph-Armand Bombardier Canada Graduate Scholarship (CGS-M), and the Department of National Defence through the Mobilizing Insights in Defence and Security (MINDS) scholarship initiative. Please note there is no grant number associated with these awards.”

I have shifted this statement to the cover letter, and added the requested statements.

Before we proceed with your manuscript, please address the following prompts

Since our data are qualitative in nature and our participants are often the only woman working in their professional community, they contain identifying and sensitive information. As a result, making our data publicly available could jeopardize the professional anonymity and safety of our participants.. However, we would consider providing an anonymized data set upon receiving a request, which can be made to our Human Research Ethics Board at researchethics@[XXXXX].

Please note that we have updated our Data Availability statement. For further details, please see our response to question 5.

5. We note that your Data Availability Statement is currently as follows: [All relevant data are within the manuscript and its Supporting Information files.]

In addition to our answer to question 4, the following response has replaced our previous statement on data availability:

Data cannot be shared publicly because of for ethical reasons, as they contain identifying and sensitive information. Data are available from the University of Waterloo Institutional Data Access / Ethics Committee (contact via researchethics@uwaterloo.ca) for researchers who meet the criteria for access to confidential data.

6. Please ensure that you include a title page within your main document. You should list all authors and all affiliations as per our author instructions and clearly indicate the corresponding author.

We have made these requested edits to the manuscript file.

7. Please update your submission to use the PLOS LaTeX template. The template and more information on our requirements for LaTeX submissions can be found at http://journals.plos.org/plosone/s/latex.

We have made this requested edit to the manuscript file.

8. Please include your tables as part of your main manuscript and remove the individual files. Please note that supplementary tables (should remain/ be uploaded) as separate "supporting information" files

Please note that we have converted what was originally a table into an in-text list.

Additional Editor Comments (if provided):

Reviewer #1: Abstract

Comment 1, Page 4, Line 1: Consider rephrasing “designed”. Emergency response work is not necessarily gendered in its design.

We thank the reviewer for drawing our attention to the wording of the abstract. Our use of the term “designed” was meant to recognize that men have represented the vast majority of individuals who worked as first responders. As a result, the systems put in place to manage and effectively utilize this human resource has been shaped by this circumstance. In referring to ‘systems’, we mean the routine ways of enabling first responders to perform their work, such as through the design of uniforms and equipment, or approaches taken toward scheduling, for instance. By opening with an acknowledgment of this history, we aim to provide context to the reader about the social norms embedded within first response delivery systems, on which present day work continues to operate, because when a dominant characteristic (such as being a man) is so deeply intertwined with the idea or archetype of a social role (such as an emergency responder), it can sometimes appear invisible.

Introduction

Comment 1: Please be consistent with your language, public safety vs. first responder.

We thank the reviewer for the suggestion and have made adjustments to use consistent terminology.

Comment 2: A definition of what a first responder is would be helpful to focus the reader on the cohorts you are specifically commenting on. There is a lot of back and forth in the introduction between FF, PO etc, so it would be help to define the groups you are presenting on.

We thank the reviewer for the suggestion. We have added the following broad definition of “first responders” and clarified the focus of this specific study:

Line 88: “Notably, we take “first response work” to include those professions that respond to and manage medical, safety, or environmental crisis situations in community settings, such as those in the fire service, police service, and paramedicine.”

Line 125: “While there are many first responder roles, including 911 dispatchers and emergency medical technicians, the scope of this project focuses on those most commonly found in the Canadian context: paramedics, firefighters, and police officers.”

Comment 3: It is worth mentioning the increased risk for adverse mental health outcomes among these groups, of which are further elevated among women. EMS and fire, specifically, are noted to have increase depression, anxiety, substance abuse, and even suicide. This speaks volumes when it comes to advocating for more research among these cohorts. Please consider including some of the citations below.

McCann-Pineo M, Keating M, McEvoy T, Schwartz M, Schwartz RM, Washko J, Wuestman E, Berkowitz J. The Female Emergency Medical Services Experience: A Mixed Methods Study. Prehospital Emergency Care. 2024 May 18;28(4):626-34.

Huang G, Chu H, Chen R, Liu D, Banda KJ, O’Brien AP, Jen HJ, Chiang KJ, Chiou JF, Chou KR. Prevalence of depression, anxiety, and stress among first responders for medical emergencies during COVID-19 pandemic: A meta-analysis. Journal of global health. 2022;12.

Stanley IH, Hom MA, Joiner TE. A systematic review of suicidal thoughts and behaviors among police officers, firefighters, EMTs, and paramedics. Clinical psychology review. 2016 Mar 1;44:25-44.

Stanley IH, Boffa JW, Smith LJ, Tran JK, Schmidt NB, Joiner TE, Vujanovic AA. Occupational stress and suicidality among firefighters: Examining the buffering role of distress tolerance. Psychiatry research. 2018 Aug 1;266:90-6.

Lebeaut A, Tran JK, Vujanovic AA. Posttraumatic stress, alcohol use severity, and alcohol use motives among firefighters: The role of anxiety sensitivity. Addictive Behaviors. 2020 Jul 1;106:106353.

We thank the reviewer for the recommendations. We have reviewed and considered them for inclusion.

Comment 4: What was the author’s justification for not including EMT’s? The literature suggest that they experience significant occupational and mental health burdens alongside their paramedic counterparts. There needs to be a firm rationale for their exclusion.

In Canada, when 911 is contacted for a medical emergency, a paramedic is typically dispatched, whereas in the US, it is more typically an EMT (Emergency Medical Technician).We have noted the limits of our focus and suggested that further research should include EMTs as well as other first responders, such as 911 dispatchers:

Line 615: “Lastly, given that we focused on the first responder groups which make up the majority of those who work in the Southern Ontario context, we were unable to investigate the experiences of women who work in first response professions beyond the three previously mentioned. As a result, and in line with the dearth of literature on the subject, future research is required to gain a sense of gender equity in other first response roles, such as Emergency Medical Technicians and 911 dispatchers.”

Comment 5: Has there been any efforts to collect the demographic distribution among the professions? What is the reason for not collecting it? Time, funding? Might be something to comment on.

We thank the reviewer for the reflection. As we described on line 130, many Canadian provinces do not collect gendered demographic data on the first response workforce. Some scholars argue that this is part of a larger and problematic landscape of aggregated provincial and national data which fail to sufficiently provide for analyses of gender, race, and other characteristics.

Methods

Comment 1: Please refer to and utilize the Consolidated Criteria for Reporting Qualitative Research (COREQ) Checklist., There are numerous items that are missing and not included in the main text.

The additional items have been added at various points to the main text.

More detail is need on the recruitment methodology: how many agencies/institutions were provided with the recruitment poster? Besides being active/on leave/retired <3 years, was there any other inclusion/exclusion criteria?

We thank the reviewer for the questions. We have added the requested information, where possible, to line 147. Because our recruitment poster was shared on social media, it remains unclear to us as to how many different agencies and institutions were provided with the recruitment poster.

Who developed the interview guide? Was it reviewed or tested prior to use?

This information has been added on line 175.

Please indicate how long each interview session was.

This information has been added on line 183.

There is no justification for the sample. Was saturation among themes met at 20? How did the authors decide to stop enrollment/interviews?

The following was added on line 191: “At twenty interviews, a repetition of narratives was beginning to occur, therefore, given the richness of the interview data and in recognition of the study’s human resource limitation, enrollment was stopped.”

Were the ZOOM sessions just audio recorded, or video as well? If video was used, please state. Further, if video was used, do you think that responses among those who opted for video/ZOOM may have been biased? Less confidentiality with video, which may have prevented participants from being 100% forthcoming.

Yes, some participants did choose to have their video on while being recorded, while others chose to keep it off - the option was provided upfront to all participants. Further details on this have been added to line 186. For those

---

## [Decision Letter · Decision Letter 1]

18 Jul 2025

PONE-D-24-42534R1
Improving Spaces for Women First Responders: A grounded theory on gender equity
PLOS ONE

Dear Dr. Gregory,

Thank you for submitting your manuscript to PLOS ONE. After careful consideration, we feel that it has merit but does not fully meet PLOS ONE’s publication criteria as it currently stands. Therefore, we invite you to submit a revised version of the manuscript that addresses the points raised during the review process.

We look forward to receiving your revised manuscript.

Kind regards,

Saravana Kumar

Academic Editor

PLOS ONE

Journal Requirements:

3. We note that some information in your manuscript text has been anonymized (e.g. ethics committee information on lines 143-144; acknowledgements). Please restore all anonymized information to your manuscript text. If you have any concerns about this request or would like to discuss this further, please contact plosone@plos.org before submitting your revised manuscript files.

Reviewers' comments:

Reviewer's Responses to Questions

**Comments to the Author**

1. If the authors have adequately addressed your comments raised in a previous round of review and you feel that this manuscript is now acceptable for publication, you may indicate that here to bypass the “Comments to the Author” section, enter your conflict of interest statement in the “Confidential to Editor” section, and submit your "Accept" recommendation.

Reviewer #2: All comments have been addressed

2. Is the manuscript technically sound, and do the data support the conclusions?

Reviewer #2: Yes

3. Has the statistical analysis been performed appropriately and rigorously? 

Reviewer #2: N/A

4. Have the authors made all data underlying the findings in their manuscript fully available?

Reviewer #2: Yes

5. Is the manuscript presented in an intelligible fashion and written in standard English?

Reviewer #2: Yes

6. Review Comments to the Author

Reviewer #2: This revision has increased the impact of this work for publication. Please note some typos in the following lines for correction:

233-234: "both" does not reference two like things in order. Revise for clarity.

379: Replace "loose" with "lose"

503: Replace "bullshit" with "[expletive]"

7. PLOS authors have the option to publish the peer review history of their article (what does this mean?). If published, this will include your full peer review and any attached files.

Reviewer #2: No

---

## [Author Response · Author response to Decision Letter 2]

31 Jul 2025

Response to Reviewers

PONE-D-24-42534 R1

Improving Spaces for Women First Responders: A grounded theory on gender equity

PLOS ONE

We thank both the editor and the two reviewers for their consideration of our manuscript. We have carefully reflected upon the generous comments provided and have outlined below our responses to the concerns that were raised. We are hopeful that our manuscript will now be considered acceptable for publication in PLOS ONE.

Journal Requirements:

We thank the editor for the recommendations. For further clarity, upon reviewing the suggested articles, we found that McCann-Pineo et al’s 2024 paper to be relevant, and it was added to the literature referenced on line 111. The other articles suggested were not relevant specifically to women’s experiences as first responders, and as a result were not included in the manuscript.

The reference list has been reviewed, and no changes were required.

3. We note that some information in your manuscript text has been anonymized (e.g. ethics committee information on lines 143-144; acknowledgements). Please restore all anonymized information to your manuscript text. If you have any concerns about this request or would like to discuss this further, please contact plosone@plos.org before submitting your revised manuscript files.

We thank the editor for flagging these missing elements. The previously anonymized parts are now included.

Reviewers' comments:

Reviewer's Responses to Questions

Comments to the Author

1. If the authors have adequately addressed your comments raised in a previous round of review and you feel that this manuscript is now acceptable for publication, you may indicate that here to bypass the “Comments to the Author” section, enter your conflict of interest statement in the “Confidential to Editor” section, and submit your "Accept" recommendation.

Reviewer #2: All comments have been addressed

2. Is the manuscript technically sound, and do the data support the conclusions?

Reviewer #2: Yes

3. Has the statistical analysis been performed appropriately and rigorously?

Reviewer #2: N/A

4. Have the authors made all data underlying the findings in their manuscript fully available?

Reviewer #2: Yes

5. Is the manuscript presented in an intelligible fashion and written in standard English?

Reviewer #2: Yes

6. Review Comments to the Author

Reviewer #2: This revision has increased the impact of this work for publication. Please note some typos in the following lines for correction:

233-234: "both" does not reference two like things in order. Revise for clarity.

379: Replace "loose" with "lose"

503: Replace "bullshit" with "[expletive]"

We thank the reviewer for these suggestions.

We have made edits to the statements on 233-234 to add clarity, which now reads as follows:

“Some participants also spoke about increasing accessibility to uniforms and equipment designed for women’s bodies. For example, the paramedic’s stair chair with longer arms and a woman’s MOLLE vest (a tactical vest worn by police officers) are designed to provide more space for breasts in their use, or, in the case of firefighting, smaller oxygen face masks that accommodate a women’s typically more narrow face shape.”

We have also made the suggested edits to lines 379 and 503.

7. PLOS authors have the option to publish the peer review history of their article (what does this mean?). If published, this will include your full peer review and any attached files.

Do you want your identity to be public for this peer review? For information about this choice, including consent withdrawal, please see our Privacy Policy.

Reviewer #2: No

---

## [Editor Report · Decision Letter 2]

7 Aug 2025

Improving Spaces for Women First Responders: A grounded theory on gender equity

PONE-D-24-42534R2

Dear Dr. Gregory,

We’re pleased to inform you that your manuscript has been judged scientifically suitable for publication and will be formally accepted for publication once it meets all outstanding technical requirements.

Kind regards,

Saravana Kumar

Academic Editor

PLOS ONE

---

## [Editor Report · Acceptance letter]

PONE-D-24-42534R2

PLOS ONE

Dear Dr. Gregory,

I'm pleased to inform you that your manuscript has been deemed suitable for publication in PLOS ONE. Congratulations! Your manuscript is now being handed over to our production team.

Kind regards,

on behalf of

Professor Saravana Kumar

Academic Editor

PLOS ONE